# Growth Features of Bi_2_Te_3_Sb_1.5_ Films on Polyimide Substrates Obtained by Pulsed Laser Deposition

**DOI:** 10.3390/ma15248993

**Published:** 2022-12-16

**Authors:** Alexander E. Shupenev, Svetlana L. Melnik, Ivan S. Korshunov, Sergey D. Karpoukhin, Stanislav G. Sazonkin, Alexander G. Grigor’yants

**Affiliations:** 1Department of Laser Technology in Engineering, Bauman Moscow State Technical University, 105005 Moscow, Russia; 2Department of Materials Science, Bauman Moscow State Technical University, 105005 Moscow, Russia; 3Scientific and Educational Center “Photonics and IR Technology”, Bauman Moscow State Technical University, 105005 Moscow, Russia

**Keywords:** pulsed laser deposition, thin films, stoichiometry, polyimide, technology, process innovation, thermoelectric effect

## Abstract

Thermoelectric materials in the form of thin films are used to create a wide variety of sensors and devices. The efficiency of these devices depends on the quality and efficiency of the thermoelectric materials obtained in the form of thin films. Earlier, we demonstrated that it is possible to obtain high-performance Bi_2_Te_3_Sb_1.5_ films less than 1 μm thick on polyimide substrates by using the PLD method, and determined optimal growth conditions. In the current work, the relationship between growth conditions and droplet fraction on the surface, microstructure, grain size, film thickness and chemical composition was studied. A power factor of 5.25 μW/cm×K^2^ was achieved with the reduction of droplet fraction on the film surface to 0.57%. The dependencies of the film thickness were studied, and the effect of the thickness on the efficiency of the material is shown. The general trend in the growth dynamics for Bi_2_Te_3_Sb_1.5_ films we obtained is the reduction of crystalline size with Pressure-Temperature (PT) criterion. The results of our work also show the possibility of a significant reduction of droplet phase with simultaneous management of crystalline features and thermoelectric efficiency of Bi_2_Te_3_Sb_1.5_ films grown on polyimide substrates by varying growth conditions.

## 1. Introduction

Thermoelectric (TE) materials can convert heat to electricity, or vice versa. Thermoelectrics have long been too inefficient to be cost effective in most applications. However, growth of interest in thermoelectrics applications began in the mid 1990s, when complex bulk materials were being explored and found that high efficiencies could indeed be obtained.

The efficiency of thermoelectric devices depends on the efficiency of thermoelectric materials used [1,2], which is described by the following equation [3,4]:
*ZT* = *T* × *α*^2^ × *σ*/*k*,(1)
where *ZT*—the dimensionless thermoelectric figure of merit; *α*, *σ*, *k*—Seebeck coefficient, electrical and thermal conductivity, respectively; *T*—the average absolute temperature. A value of power factor (PF = *α^2^* × *σ*) is also commonly used for rapid characterization of thin film thermoelectric efficiency.

A basic structure unit of a thermoelectric device is a thermopile that consists of a pair of connected p- and n- type semiconductors. Bi_2_Te_3_ was first investigated as a material of great thermoelectric promise in the 1950s [5]. Bi_2_Te_3_Sb_1.5_ is now among the most promising p-type materials [6,7] for the room temperature range because of its high *α,* about 210 μV/K^−1^ and ZT about 1.1, which is one of highest among bulk p-type thermoelectric materials known [7]. As a substrate, the polymer class of materials is promising [8], especially polyimide materials similar to Kapton [9,10]. The thermal expansion coefficient of polyimide is 12 × 10^−6^ K^−1^ is approximate to that of bismuth telluride coefficient (20×10^−6^ K^−1^), which should have a favorable effect on film stresses and adhesion. The advantages of polyimide include low (0.12 W·m/K) thermal conductivity, temperature resistance up to 400 °C, market availability, chemical and radiation resistance.

Modern applications of thin film thermoelectrics are wide and promising. Thermoelectric materials in the form of thin films are used to create thermo-elements in radiometry [11,12], microcalorimetry [13,14,15], in nondispersive infrared (NDIR) gas sensors [16,17], and in microgenerators [18,19]. Flexible lgTEGs are of particular interest for modern self-sustained monitoring systems [20,21]. Future low-gradient thermoelectric generators (lgTEG) that use waste thermal energy from the human body or heated structural elements for electricity generation can become a convenient compact source of energy for low-power electronics [22,23,24,25]. The successful mass introduction of lgTEG today rests mainly on solving of the problems of technological implementation of the design, and in particular on the production of thermoelectric materials in the form of thin films. Two main technological challenges can be identified for thin film thermoelectric devices such as lgTEG implementation: the use of flexible substrates for thermoelectric layers and the use of highly efficient thermoelectric materials [26,27,28,29].

Obtaining Bi2Te3Sb1.5 in the form of thin films less than 1 μm thick by physical vacuum deposition methods is complicated by the following reasons:

Significant difference in partial pressures of the elements included in the formula leads to an inhomogeneous expansion of the target material evaporated under vacuum.Abundant re-evaporation of volatile elements at high temperatures of the substrate.Low sticking coefficient of Te (<0.6) at substrate temperatures below 300 °C [7].

To overcome these limitations, the method of pulsed laser deposition may be promising. The high energies of the expansion particles (from 1 to 100 eV) of the plasma level out the difference in partial pressures [30], which leads to a congruent transfer of the target material to the substrate. This method has been implemented to obtain thin films of bismuth telluride on solid substrates [31,32]. The use of PLD method to obtain thin Bi2Te3Sb1.5 films on flexible polyimide non-oriented substrates may provide great opportunities for further scientific research and industrial applications.

In this study, p-type Bi2Te3Sb1.5 thermoelectric materials in the form of thin films were synthesized on polyimide substrates by pulsed laser deposition method. The influence of growth conditions on chemical, morphological and thermoelectric features of thin films was explored.

## 2. Materials and Methods

To obtain thin films, we used the method of pulsed laser deposition, where the radiation source was a CompexPro102F KrF excimer laser with a pulse duration of 30 ns and a wavelength of 248 nm. The films were deposited on substrates at various temperatures from 25 to 500 °C, pressures from 1 × 10^–7^ to 1 Torr in an inert gas medium (Ar 99.99%), and a distance from the target to the substrate from 70 to 110 mm. To ensure the homogeneity of the layers, the rotation of the target and substrate was used, as well as the motion of the beam along the target surface. The scheme of the process of pulsed laser deposition is shown in Figure 1.

Cylindrical single-crystal Bi_2_Te_3_Sb_1.5_ ingots were used as targets. According to the passport data, the material has a Seebeck coefficient of 205 ± 5 µV·K^−1^, an electrical conductivity of 1050 ± 150 Ω^−1^·cm^−1^ and a specific thermal conductivity of about 1.4 ± 0.05 W·m^−1^·K^−1^. The film thickness was about 300 nm. The target surface was polished before each technological process to enhance droplet-free film surface state. The substrates were polyimide material similar to Kapton, 100 µm thick and with the following characteristics: tensile strength from 135 to 1350 MPa, specific volume electrical resistance 1×10^15^ Ω·cm, dielectric constant 4 × 10^6^ N, dielectric loss tangent 1 × 10^−3^ Ω cm.

The thermoelectric properties of the films were measured on a laboratory measuring bench by the probe method in linear geometry. Since the nature of the distribution of thermoelectric properties over the area of the substrates can be inhomogeneous, the bench design allows measurements of the values *α* and *σ* in one area. The measuring head includes six probes made of closely spaced copper pointed needles. Measurements of the Seebeck coefficient and electrical conductivity are carried out sequentially.

Two probes are used to measure the Seebeck coefficient by standard technique utilizing heating one probe and measuring the voltage drop between probes. The central group of probes is used to measure the sheet resistivity by four-point probe method with the use of corresponding correction factor for thin film case [33]. The film thickness required to use this technique in the measured area was taken from a thickness distribution model obtained earlier on the basis of experimental measurements of the image thicknesses using the Ntegra Spectra scanning probe microscope and the KLA Tencor P-17 profilometer. 

Studies of the chemical composition of the films were carried out using the Tescan Vega II LMH scanning electron microscope with the Oxford INCA 350 energy dispersion analyzer of chemical composition. Image analysis when calculating the volume of the teardrop fraction and grain size was carried out using the ImageJ program. To increase the contrast at the drop/substrate boundary, a Bandpass filter implemented in the built-in fast Fourier transform process was used. Next, the Threshold filter converted the image into binary with separation into droplets and a substrate. The analysis of the volume of the teardrop fraction included measuring the area of each individual drop, including at the borders of the image, followed by recalculation into the total occupied area.

## 3. Results

In this work, two series of experiments were carried out to identify the technological patterns of the appearance of a droplet fraction, and to study the dependence of thermoelectric properties and crystal size on the conditions for obtaining a thin film coating. In this case, the variable technological parameters were temperature, pressure, and the distance between the target and the substrate. According to the results of the analysis of a series of samples, a significant influence of the technological parameters of the process of obtaining films on their morphological features can be seen.

### 3.1. Study of the Droplet Fraction

The appearance of a droplet fraction on the surface of thin films obtained by PVD methods is a well-known problem [34]. The presence of a drop phase on the surface of films can have a significant negative effect in solving problems of nano-structuring of thermoelectric materials [35,36] in order to increase the thermoelectric efficiency. The reason for the formation of a droplet phase in the case of using the PLD method is a complex of various phenomena [35] closely related to the parameters of laser treatment and growth conditions.

Figure 2 shows the results of studying the drop fraction on the film surface. The experimental plan (shown in Figure 2a) was drawn up for different modes: temperatures (T): 250, 400 and 500 °C; and distances between the target and the substrate (H): 70, 110 and 150 mm at a fixed pressure P = 1 Torr. The selected samples 7, 5, 3 attract attention with a clear trend towards a decrease in the volume of the droplet fraction (Figure 2b) and the average droplet size. Figure 2c shows an example of image processing by the ImageJ package to calculate the number of drops is given. From left to right: enlarged image of the deposited film, circled contours of drops, hypothetical topology of the film.

### 3.2. Study of the Films Morphology

The study of the microstructure consisted of two stages. To study the dependence of grain size on laser deposition modes, an experimental plan was drawn up and is shown in Figure 3a, in which the following parameters were changed: the pressure (P) in the vacuum chamber was: 0.0001, 0.01, 0.1 and 1 Torr; temperature (T): 25, 200, 350, 500 °C. Pictures of the surface of the selected samples 1, 6, 11, 16 are shown in Figure 3b. The microstructure of the selected samples differs significantly in grain size. Sample 1 does not show an island structure without formed grains, while the rest of the samples show a grain structure with distinct boundaries in samples 6 and 16.

### 3.3. Study of the Films Physical Properties

For the studied samples, the chemical composition was additionally studied using an energy-dispersed analyzer, which can be seen in Figure 4. The initial composition of the target was Bi = 9.53%, Te = 59.04%, 34.43%. A good stoichiometry of the deposited film with a thickness of up to 300 nm is observed: on average, the percentage of antimony deviates ΔSb = 1% from the initial one; tellurium ΔTe = 2.8%; and bismuth ΔBi = 3.6%. The chemical composition of the droplets differs from the composition of the film by a lower content of tellurium with an average deviation of ΔTe up to 6%, and an increased content of bismuth ΔBi with an average deviation of 4.6%.

## 4. Discussion

### 4.1. Analysis of the Droplet Fraction and Films Morphology

The graph in Figure 5 shows the ratio of the number of drops per unit area versus temperature T and the distance between the target and the substrate H. There are three fundamental processes that cause droplet formation during PLD: subsurface boiling, shock wave recoil pressure expulsion and exfoliation. Thus, target-to-substrate distance should have a high influence on the amount of droplet phase on films surface due to it’s mechanical nature. We observe a decrease in the droplet fraction from 8.58 to 0.57% with temperature and target-to-substrate distance increase.

Figure 6 shows a plot of the main properties of the film: Seebeck coefficient α, electrical resistance R, and power factor PF on temperature T and pressure P at a fixed distance between the target and the substrate H = 70 mm. The electrical resistance decreases for the selected series of samples, which corresponds to the previously obtained experimental data [32]. Presumably, this is explained by an increase in the size of crystals and film defects, which leads to an increase in carrier mobility and carrier concentration, which increases the electrical conductivity [37]. Initially, the slowly increasing Seebeck coefficient increases abruptly for sample 11, and then decreases for sample 16. This phenomenon is presumably associated with an increase in the internal concentration of electron carriers due to thermal excitation [38].

Sample 11 in Figure 3 turned out to be optimal for current research, which combines an increased Seebeck coefficient and low resistance, which leads to a maximum electrical power factor of 5.25 μW/cm×K^2^. With a further increase in temperature to 500 °C and pressure to 1 Torr, the resistance continues to decrease, but the Seebeck coefficient also decreases, which leads to a decrease in the power factor. This mode is close to the studies [25], where at a temperature of 400 °C the maximum power factor PF = 11.45 × 10^−4^ W/m·K^2^ was achieved. With an increase in the grain size to 0.6 μm, α increases, the resistance R remains practically unchanged, which leads to an improvement of PF. After a grain size of 0.45 µm, the growth of α slows down, and the resistance R is at a minimum level, which gives the highest PF. After a grain size of 0.6 μm, α begins to decrease, while the resistance R highly increases, which leads to a serious decrease of PF.

The distribution of the average grain size for each sample is shown in Figure 7. The average grain size was obtained from histograms. For sample 6 with a relative frequency of 26%, the grain size is 0.23 µm; for sample 11, the relative frequency was 25% for grains with a size of 0.14 µm. Similar grain sizes were obtained in [39] with similar processing parameters. The distribution of relative grain size frequency in sample 16 (see Figure 7) has two peaks: with a relative frequency of 55%, grains with a size of 0.2 μm are found in it, and with a relative frequency of up to 4%, grains with sizes ranging from 1 to 10 μm are observed. The excessive grain growth combined with fine grains for sample 16 may be due to the secondary recrystallization process occurring at high temperatures. A similar picture is observed in the studies [40], however, the temperatures there were lower, which is probably due to the technological features of the film growth process.

### 4.2. Analysis of the Chemical Composition and Films Thikness

At a film thickness of up to 100 nm, the presence of oxygen and carbon is detected in the chemical composition, which may be due to the discontinuity of the film. Another assumption is that the distance between the target and the substrate is too large, which affects the appearance of defects due to the interaction of laser ablation products with gas atoms in the chamber, the number of which is proportional to pressure. The smallest deviation from stoichiometry is maintained under the following modes: pressure 0.1 Torr, temperature 250 °C, distance between the target and the substrate 70 mm, and is ΔSb = 0.5%, ΔTe = 1.45% and ΔBi = 1%, and the largest occurs when the pressure rises to 1 Torr and the above regimes are maintained and amounts to ΔSb = 4.65%, ΔTe = 7.8% and ΔBi = 12.45%.

One of the significant influences on the thermoelectric properties of the film is its thickness. Figure 8 shows the dependences of the thickness on the main technological parameters of the process, such as the substrate temperature, the pressure in the chamber, and the distance between the substrate and the target, are shown.

A combination of technological parameters is observed in which the same duration of the deposition process and, as a result, the same amount of evaporated material gives a greater film thickness. At a distance of 90 to 120, a temperature of 450 °C and a pressure of 1 Torr, the maximum film thickness reaches 151 nm. However, it can be noted that the ratio of distance to pressure has a lesser effect than the ratio of temperature to pressure, which have a greater effect on the amount of material on the substrate and have clear peak regions. At high pressure and low temperature, we observe a significant reduction in deposited material, as well as at low pressure and high temperature.

Figure 9 shows a plot of the main properties of the film: the Seebeck coefficient α, electrical resistance R, and power factor PF on the film thickness. As can be seen from the graph, there is a clear optimum in the thickness range from 120 to 140 nm, before and after it the film has too high resistance.

## 5. Conclusions

The morphological features of samples of Bi_2_Te_3_Sb_1.5_ thermoelectric films obtained by pulsed laser deposition on polyimide substrates were studied in this work. We found that the thermoelectric efficiency of Bi_2_Te_3_Sb_1.5_ films is highly dependent on morphological structure, and it may be managed by the growth conditions of the pulsed laser deposition method. We also observed that the best result is achieved with a small deviation of the film composition from the target composition, which can be varied by combinations of film growth conditions.

In the work, the energy PT-criterion was chosen as having the most significant effect on the structural and physical parameters of the resulting films. Structural studies of the samples showed that with an increase in the parameter PT, the characteristic grain size increases. The sample with the highest electric power factor of 5.25 μW/cm×K^2^ has grains with a characteristic size of 0.14 μm, with a relative frequency of 25%. The amount of the droplet phase on the surface observed to be inversely proportional to the selected energy PT criterion and for the investigated range of technological parameters varied from 8.58 to 0.57%.

The general trends in the growth dynamics of the crystal structure of Bi_2_Te_3_Sb_1.5_ films are similar to those observed in the literature [41,42,43] when using other methods of physical deposition. A positive feature of the PLD method is the possibility of congruent material transfer, but a negative feature is the formation of a droplet phase on the surface [44,45]. The results of current research show the possibility of a significant reduction in the droplet phase with simultaneous optimization of the crystallite size and efficiency of Bi_2_Te_3_Sb_1.5_ films on polyimide substrates by choosing the optimal growth conditions.

## Figures and Tables

**Figure 1 materials-15-08993-f001:**
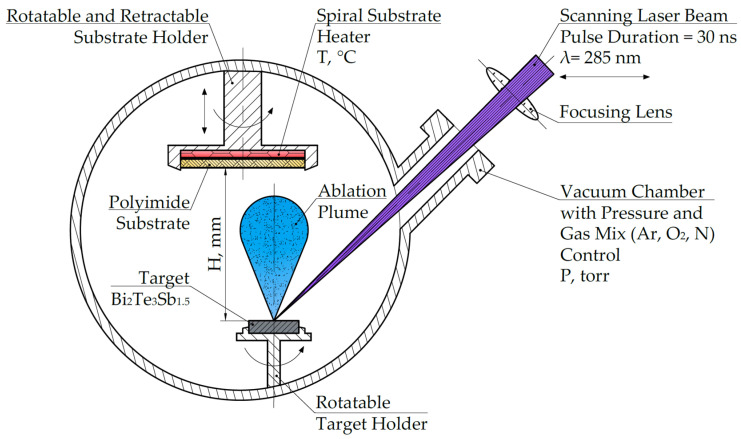
Scheme of the process of pulsed laser deposition.

**Figure 2 materials-15-08993-f002:**
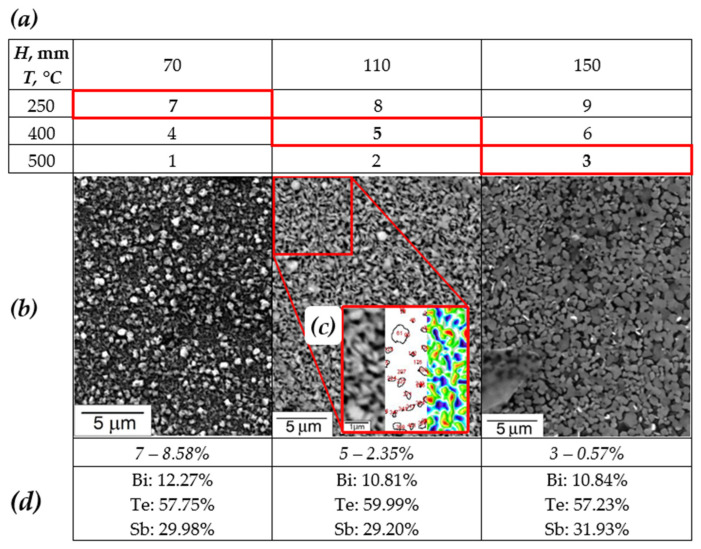
The results of the study of the droplet fraction on the surface of thin films: (**a**) the plan of the experiment where the red borders of the cell indicate the samples selected for analysis; (**b**) images of the microstructure of the samples; (**c**) example of image processing by the ImageJ package; (**d**) EDS results for selected samples.

**Figure 3 materials-15-08993-f003:**
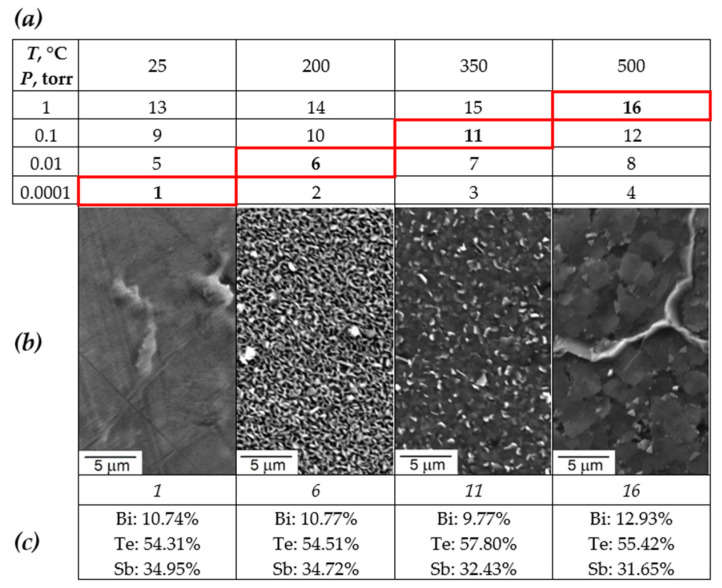
The results of the study of the microstructure of thin films: (**a**) the plan of the experiment where the red borders of the cell indicate the samples selected for analysis; (**b**) images of the microstructure of the samples. (**c**) EDS results for selected samples.

**Figure 4 materials-15-08993-f004:**
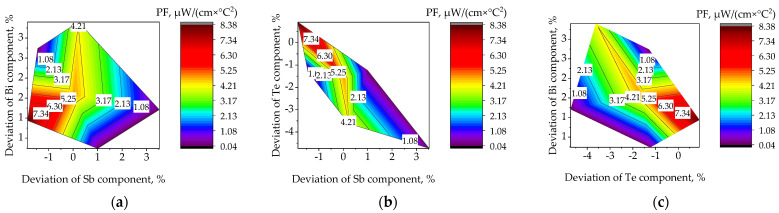
The results of the study of the chemical composition of thin films: plots of thermoelectric power factor, PF, on the deviation of the stoichiometric composition for the main components, Bi, Te, Sb: (**a**) Bi and Sb; (**b**) Te and Sb; (**c**) Bi and Te.

**Figure 5 materials-15-08993-f005:**
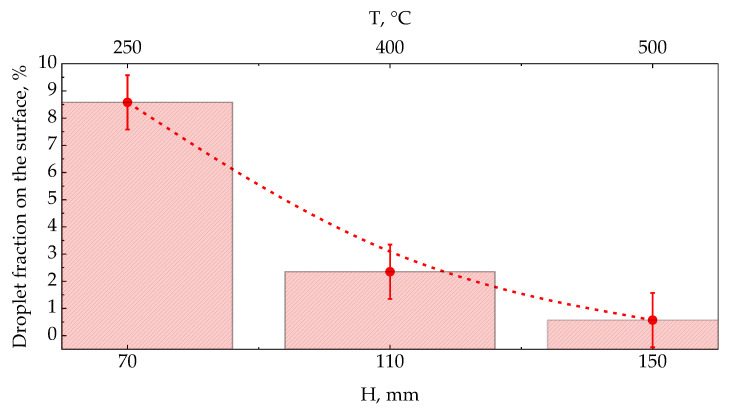
Graph of the ratio of the number of drops per unit area of temperature, T, and the distance between the substrate and the target, N, at a fixed pressure value in the chamber, P = 1 torr.

**Figure 6 materials-15-08993-f006:**
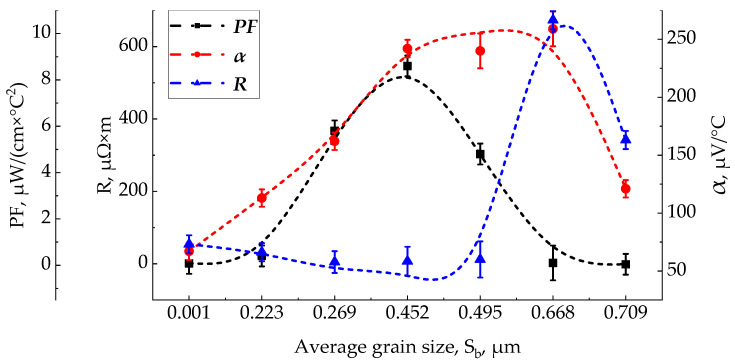
Plot of basic properties, α, R, PF, versus average grain size, S_b_.

**Figure 7 materials-15-08993-f007:**
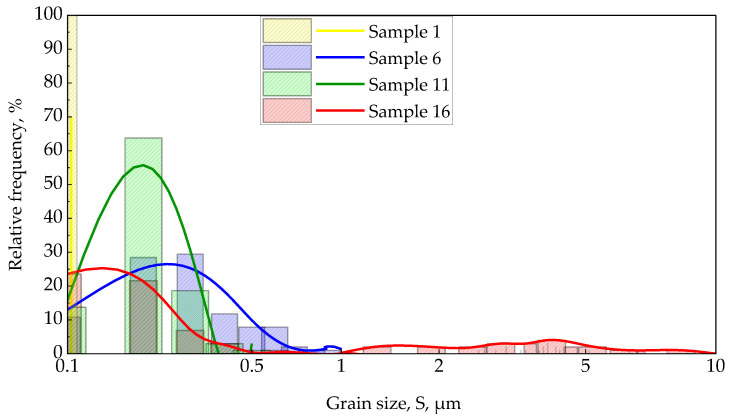
Plot of relative frequency versus grain size, S.

**Figure 8 materials-15-08993-f008:**
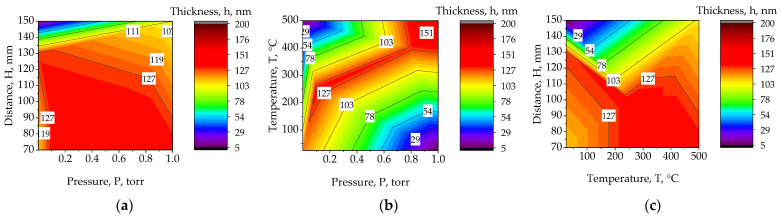
Plots of the dependence of the thin film thickness, h, on the main parameters of the technological process, T, P, H: (**a**) H and P; (**b**) T and P; (**c**) H and T.

**Figure 9 materials-15-08993-f009:**
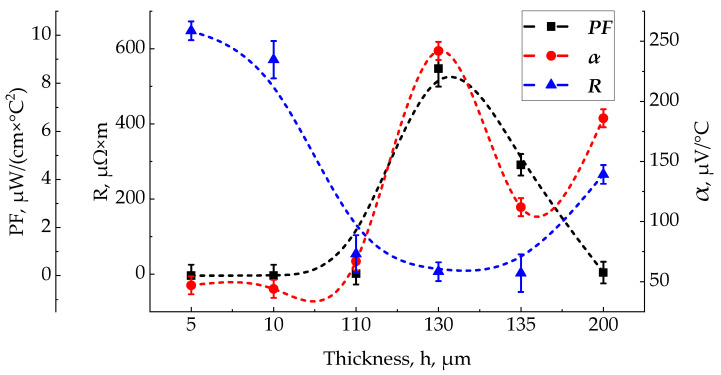
Plot of the main properties, α, R, PF versus the thickness of a film, h.

## Data Availability

Not applicable.

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
