# Peer review of "Growth Features of Bi2Te3Sb1.5 Films on Polyimide Substrates Obtained by Pulsed Laser Deposition"

_materials, 2022, doi:10.3390/ma15248993_

Round 1

Reviewer 1 Report

The manuscript titled “Growth features of Bi2Te3Sb1.5 films on polyimide substrates obtained by pulsed laser deposition” samples of Bi2Te3Sb1.5 thermoelectric films were obtained by pulsed laser deposition on polyimide substrates. The sample with 237 the highest electric power factor of 5.25 μW/cm∙K has grains with a characteristic size of 0.14 μm with a relative frequency of 0.25. The amount of the droplet phase on the surface turned out to be inversely proportional to the selected energy PT criterion. The article is well written. I have only one note mentioned below.

The authors mentioned that “The general trends in the growth dynamics of the crystal structure of Bi2Te3Sb1.5 films 242 are similar to those observed in the literature when using other methods of physical deposition.” You should state some examples in comparisons to your results. You may put is as a table to be clear to the readers.

Author Response

Thank you very much for your interest in our article! We have made changes in accordance with your comments.

We have added references we intended at "...the crystal structure of Bi2Te3Sb1.5 films are similar to those observed in the literature..." 

Reviewer 2 Report

Dear Editor,

About the manuscript : materials-2080701

Title: Growth features of Bi2Te3Sb1.5 films on polyimide substrates obtained by pulsed laser deposition.

Authors: A. E. Shupenev et al.

In this work, thermoelectric materials in the form of thin films have been used to create a wide variety of sensors and devices. In particular,  the authors indicated that high-performance can be obtained by Bi2Te3Sb1.5 using PLD method with optimal growth conditions. The results show that the possibility of a significant reduction in the droplet phase with simultaneous management of crystallite features and efficiency of Bi2Te3Sb1.5 films on polyimide substrates by choosing optimal growth conditions.

My recommendation is that the paper should be revised: Minor Revision.

1) Introduction part should contained research papers/latest/related or research results should be cited to improve scientific quality of manuscript such as materials in the form of thin films,  for sensors, waveguide, telecommunication,... applied in physics see the following works:

Infrared Phys. Technol. vol.89 (2018) p.218.https://doi.org/10.1016/j.infrared.2018.01.009

Eur. Phys. J. Plus vol.137 (2022) p-1137: https://doi.org/10.1140/epjp/s13360-022-03351-w

Micro and Nanostructures, vol.171 (2022) p-207417: https://doi.org/10.1016/j.micrna.2022.207417

2) The basic properties, a, R, PF, versus average grain size need to be more explained.

3) Before concluding the manuscript, it would be nice to add highlight and the advantages the proposed work.

4) Reference part needs to be improved by a proper extension as per above mentioned suggestions.

Author Response

The team of authors would like to thank the referee for a careful reading of the article. As recommended, we have made the following changes:

1,4) We have added many new actual references to Introduction part on latest and related research results. We added some remarkable works such as Physica B 634 (2022) 413636 DOI:10.1016/j.physb.2021.413636 in reference part.

2) We have added more details to the description of the influence of the average grain size on the main properties, based on the obtained dependence.

3) We highlighted advantages in conclusions part.

Reviewer 3 Report

  1. The abstract is not showing the exact findings and output and what the authors want to show here in short. Authors have to give more time and rewrite the abstract.
  2. The introduction is poorly written by the authors and needs to elaborate more as per subject and materials advantages.
  3. Page 2, line 74: What do you mean here, “The films were deposited at different substrate…………”
  4. EDS results need to be added and explain in the manuscript.
  5. Section 4.1. Analysis of the droplet fraction and film morphology needs to elaborate in detail.
  6. What authors want to justify on the basis of Figure 8 have to discuss in detail.
  7. The conclusion needs to make more specific. It is too gigantic.
  8. Polish the grammar once.

Author Response

Thank you so much for such a large and high-quality review!

We have prepared the answers and additions in accordance with your comments:

1) We gave more time for our abstract and highlighted some of our findings of our research.

2) We gave more time for our Introduction part. Subject and materials advantages were described more. We have also added new references in Introduction part on latest and related research results.

3) We corrected the misspelling in line 74.

4) EDS results were added to the tables with samples in results part, as well as the initial composition of the target to compare with.

5) In Section 4.1 we added more details in accordance with figures.

6) Above Figure 8 we have added more details.

7) Conclusions were shortened and made more specific.

8) Grammar and spelling were polished.

Round 2

Reviewer 3 Report

Minor changes need in the manuscript before acceptance.

1. Introduction has scope to improve it really not showing exact intro of the work.

2. Authors can see the the literature Journal of Membrane Science Volume 531, 2017, Pages 77-85 and Journal of Environmental Chemical Engineering Volume 9, 2021, 104774 to improve it about the coatings.

3. y-axis unit is required for Fig. 7. 

Author Response

Thank you for your comments! We took all your suggestions into consideration and made corrections as follows:

1,2) We've improved the Introduction part and restructured it.

Now its structure follows new logic: introduction to thermoelectric materials > main properties of TE materials > why we chose Bi2Te3Sb1.5  for current research > potencial applications > technological challenges > the scope of this work

3) The mistake in Figure 7 was corrected by specifying units of the y-axis settings in %. The results associated with the Figure 7 are now described accordingly in %